# Effects of the Inclusion of Dietary Bitter Gourd (*Momordica charantia*) on the Performance and Carcass Characteristics of Pigs: Potential Application in the Feed Chain

**DOI:** 10.3390/ani13132159

**Published:** 2023-06-30

**Authors:** Xiaonan Guan, Regiane R. Santos, Sietse J. Koopmans, Francesc Molist

**Affiliations:** 1Department of Research & Development, Schothorst Feed Research, 8200 AM Lelystad, The Netherlands; rsantos@schothorst.nl (R.R.S.); fmolist@schothorst.nl (F.M.); 2Wageningen Livestock Research, Wageningen University & Research, 6700 AH Wageningen, The Netherlands; sietsejan.koopmans@wur.nl

**Keywords:** *Momordica charantia*, bitter gourd, production performance, carcass quality, pigs

## Abstract

**Simple Summary:**

One of the goals of the Feed Sustainability Charter is the inclusion of a circular feed concept in animal production. Although it is still not possible to replace traditional feed production, the supplementation of diets with alternative ingredients able to maintain animal production appears to be a promising step. This study investigated the effects of supplementing a commercial diet with leftovers (stems and leaves) of bitter gourd (6.5 or 13 g/kg) on the performance, carcass characteristics, and serum parameters of growing-finishing pigs. None of the tested inclusion levels of bitter gourd affected pig production, mortality, or carcass quality, as well as serum levels of urea, insulin, or leptin. In conclusion, bitter gourd leftovers can be included in the diet of growing-finishing pigs.

**Abstract:**

The objective of this study was to determine the effect of bitter gourd (BG) leftovers (stems and leaves) as an alternative dietary ingredient on pig performance, carcass characteristics, serum parameters (urea, insulin, and leptin levels), and faecal consistency. Healthy Tempo × Great Yorkshire and Landrace pigs (N = 240; 120 gilts and 120 boars) weighing 25.8 kg (9–10 weeks of age) were randomly assigned to three treatments (eight pens per treatment; each pen with five gilts and five boars). The three treatments consisted of a non-supplemented commercial diet (control; CON) and a CON diet supplemented with 6.5 g/kg BG (BG1) or 13 g/kg BG (BG2). Pigs were fed the experimental diets until slaughter (120 kg body weight; BW). Feed intake was recorded daily and calculated for each experimental phase (i.e., days 0–36, days 36–66, days 66–98, and the overall experimental period). Average daily feed intake (ADFI), average daily gain (ADG), and feed conversion ratio (FCR) were calculated. The frequencies of visiting the feed station and of feeding were recorded daily. Faecal scores (FS) for consistency were measured per pen twice weekly. On the day of slaughter, two pigs per pen (one male and one female) were randomly selected for the measurement of muscle thickness and blood collection. At the slaughterhouse, carcass weight, dressing percentage, back fat thickness, muscle depth, and lean meat percentage were recorded. Data were analysed using ANOVA, with the pen as the experimental unit. Diets BG1 or BG2 did not affect the performance of the pigs, except for a significant decrease in the ADG of the pigs fed the BG2 diet in the feeding period of 50–80 kg. However, no differences in performance were observed in the overall experimental period. Faecal scores, carcass quality, and serum levels of urea, insulin, and leptin were also not affected by the diet. In summary, leftovers (stems and leaves) of BG can be successfully added to the diet of growing-finishing pigs without interfering with performance and carcass characteristics.

## 1. Introduction

Animal feeding, especially in the pig industry, has an impact on the environment and on the availability of nutrient sources used to produce human diets. According to Mottet et al. [1], 40% of the global arable land is dedicated to produce animal feed (for monogastrics and ruminants), generating a food-feed competition for cultivated land. From the complete global production, approximately 33% is lost per year [2], and these losses may occur during primary production, processing and packaging, retail, and distribution, among other steps. Generally, vegetable losses are high and, in some cases, 41% of the vegetables are lost during the removal of external leaves and core [3]. Instead of using the leftovers as biomass-waste, some vegetable leaves and stems can be transformed in more valuable ingredients to be added to animal feed. According to the United States Environmental Protection Agency [4] pyramid, leftovers should be redistributed to feed animals in a higher priority than for biofuel production or composting [4,5]. The incorporation of alternative ingredients may support an eco-friendly feed production, which can be performed by feeding these losses to pigs, thereby respecting global agriculture limitations [6]. Importantly, the partial replacement of conventional ingredients in animal diets should not negatively impact animal performance and carcass value [7]. Apart from the nutrients present in these possible by-products, they also may have some positive effects on animal health and subsequent productivity.

Bitter gourd (BG, *Momordica charantia*) is a vegetable used as functional food to treat metabolic diseases in humans. It also is a rich source of bioactive molecules with anti-hyperglycaemic, immunomodulatory, antioxidant, hepatoprotective, and anti-inflammatory properties, among others [8]. The anti-hyperglycaemic activity of BG is related to its ability to improve insulin sensitivity and signalling in obese or pre-diabetic laboratory animals [9]. The inclusion levels of BG in the diets of these animals ranged from 10% to 15% in rats [10,11]. In rats fed a high-fat diet, BG at an inclusion level of up to 1.5% improved insulin resistance and lowered serum leptin levels [12]. Exposure up to 4 g/kg body weight (bw) extract of BG fruit via intraperitoneal administration did not negatively impact kidney function in mice, but chronic exposure to 0.5 g/kg bw/day resulted in nephrotoxic effects, based on increased serum urea levels [13]. In rats, the toxicity of bitter gourd extract after oral administration was observed only at doses of 2 g/kg bw [14]. Considering its bitter taste and the risk of impacting nutrient levels at high inclusion levels, BG stems and leaves should be added at lower concentrations to mimic practical conditions and to avoid impaired palatability. In the pig industry, obesity is not an issue, but improved protein metabolism, induced by increased insulin signalling, may be beneficial. Generally, the inclusion levels of ingredients are calculated to obtain optimal animal performance, which means an efficient use of the agricultural land to produce animal protein. Based on a previous study, pigs with increased insulin plasma levels have an improved use of amino acids such as lysine, tryptophan, and arginine [15]. Furthermore, insulin allows the use of amino acids for muscle protein synthesis [16], suggesting that it will support the growth of pigs. Hence, we hypothesised that supplementing a commercial diet with BG leftovers (stems and leaves) is a strategy to apply the principles of a circular economy in the pig industry without impairing animal production performance or even improving the feed conversion ratio. For this, growing-finisher pigs were fed commercial diets either supplemented or not with 6.5 or 13 g/kg dried BG leftovers. Moreover, to investigate any possible negative impacts of BG on palatability, the daily feed intake behaviour was monitored. Additionally, on the day of slaughter, serum levels of urea, leptin, and insulin, as well as carcass characteristics, were evaluated.

## 2. Materials and Methods

### 2.1. Animals and Housing

In total, 240 healthy pigs, 120 gilts and 120 entire boars (Tempo × Great Yorkshire and Landrace), with an average body weight (BW) of 25.8 ± 2.0 kg and an average age of 9–10 weeks, entered the experiment. Pigs were randomly accolated to replicates based on their weight. They were housed in groups of 10, with 5 gilts and 5 boars per pen (2.75 × 4.45 m; 12.24 m^2^ and 1.22 m^2^/pig), in the climate-controlled individual feeding stations in the Schothorst Feed Research facility. Room temperature and relative humidity were recorded daily.

### 2.2. Processing of the Bitter Gourd Leftovers

Bitter gourd was cultivated in a greenhouse and obtained from Fresh Farma, Bleiswijk, The Netherlands. Stems and leaves of this cultivar were prepared in a dried powdered form. For this, fresh clean material was cut into pieces (thick: <5 mm) and placed in an air-vented oven at 60 °C for three days until the weight of the material remained constant. The material was turned twice daily to improve drying. The dried material was milled to powder and sealed in plastic bags. Material was stored in a dark room at room temperature prior to use.

### 2.3. Diets and Experimental Design

Diets were prepared as pellets and formulated free of antimicrobial growth promoters or feed additives with antimicrobial effects. Pigs were fed according to a 3-phase feeding scheme: starter (25–50 kg BW), grower (50–80 kg BW), and finisher (80–120 kg BW) phases. The nutrient content of the basal diets met the minimum levels according to the recommendations for pig diets by the CVB (Centraal Veevoeder Bureau; Dutch Feed Table), as shown in Appendix A. Experimental diets were produced by double mixing. First, a large amount of basal diet was made, which was split into three portions. The first portions for each feeding phase were regarded as the control (CON) diets. To the second portions of basal diet, 6.25, 6.8, and 6.5 g/kg of BG (stems and leaves) were added, resulting in diet BG1 in each feeding phase. To the third portions, 12.5, 13.6, and 13.0 g/kg of BG were added, which resulted in diet BG2 in each feeding phase. The experiment was performed in a completely randomised block design with three treatments and eight replicates, as described in Table 1. Pigs from the CON group were fed a commercial diet. Pigs from BG1 and BG2 groups were fed the CON diet supplemented with different levels of BG, depending on the feeding period.

### 2.4. Production Performance

Individual pig BW was recorded at start of the trial (circa 25 kg; d0 of the trial), at days 39 and 66, and at first delivery to the slaughterhouse. Average daily feed intake (ADFI, g/pig), average daily gain (ADG, g/pig), and feed conversion ratio (FCR) were calculated on a pig basis for each experimental phase (i.e., days 0–36, days 36–66, days 66–98) and for the total grower-finisher period (i.e., days 0–98). Pigs were monitored daily for general health.

### 2.5. Faecal Consistency

In the grower-finisher rooms, faecal consistency per pen was measured twice weekly (i.e., each Tuesday and Friday) on an 8-point faecal score (FS) scale from severe water-thin diarrhoea to hard, dry, and lumpy faeces (score 6 = normal faecal consistency) [17]. Figure 1 depicts representative images of faeces at different score levels. The faecal consistency score was averaged for each experimental phase.

### 2.6. Carcass Characteristics

Individual slaughter data were obtained for all grower-finisher pigs in the trial, except for the pigs that had died. The following carcass characteristics were determined: carcass weight, dressing percentage, back fat thickness, muscle depth, and lean meat percentage. Additionally, on the day of slaughter, two pigs per pen (one male and one female) were randomly selected for the measurement of muscle thickness. Muscle and fat thickness were recorded for each individual pig using a Capture Gras-Maigre (CGM; Sydel, France), i.e., at the third to fourth from last rib position. Lean meat percentage per pig was calculated according to the following equation: 66.86 + 0.0207 × muscle thickness (mm) − 0.6549 × backfat thickness (mm) [18]. Dressing percentage was calculated by using the carcass weight divided by the slaughter body weight.

### 2.7. Blood Parameters

On the day of slaughter, the last day of the trial, two pigs per pen (one male and one female) were randomly selected for blood collection. Blood was collected after overnight (8 h) fasting to avoid feed intake interfering with insulin measurements. The blood sample was collected from the jugular vein using 5 mL Vacutainer tubes. The sampled blood was then submitted to serum harvesting via centrifugation at 1500× *g* for 10 min at 4 °C room temperature and stored frozen (−20 °C) in 1 mL vials until analysis. Serum levels of porcine urea (mmol/L) were determined using a Cobas 8000 System (Roche Diagnostics Corporation, Indianapolis, IN, USA). Serum levels of porcine insulin (µU/mL) were determined using an ADVIA Centaur System (Siemens, Den Haag, The Netherlands). Serum levels of porcine leptin (ng/mL) were determined using an assay kit from MyBiosource Inc. (San Diego, CA, USA), coded as MBS703419, and absorbance was measured at a wavelength of 450 nm (plate reader Infinite^®^ 200 Pro, Tecan, Männedorf, Switzerland).

### 2.8. Feed Intake Frequency

The experimental facilities were equipped with individual feeding stations (INSENTEC, Marknesse, The Netherlands) that registered the individual feed intake of group-housed pigs. All pigs had ear tags with unique numbering; therefore, individual feed intake records were available for all pigs for each day of the experiment.

### 2.9. Statistical Analysis

Statistical analysis was carried out with GenStat^®^ for Windows (21st edition; VSN International, Hemel Hempstead, UK). All parameters were analysed using analysis of variance (ANOVA) with Fisher’s least significant difference (LSD) test to compare treatment means. The pen was the experimental unit for all performance data, and the pig was the experimental unit for carcass, muscle thickness, and serum analyses. Given the factorial design, the statistical model used to analyse the data was:Y = µ + blocki + Dietj + Genderk + Diet*Genderjk + eijk

In which:

Y = Response parameter;µ = General mean;Blockj = Effect of block (i = 1…6);Dietj = Effect of Diet (j = 1…3);Genderk = Effect of Gender (k = 1, 2);Diet*Genderjk = Effect of the interactions between Diet and Gender;Errorijk = Error term.Treatment means were compared using least significant difference (LSD). Values with *p* ≤ 0.05 were considered statistically significant.

## 3. Results

### 3.1. Production Performance

The average body weight of the pigs was 25.8 ± 2.0 kg at day 0, 57.3 ± 9.8 kg at day 39, 82.8 ± 17.9 kg at day 66, and 120.5 ± 30.0 kg at slaughter. Average days to slaughter/experimental period was 98.4 ± 0.26 days. The overall ADG was 962 ± 154 g/pig, overall ADFI was 2.277 ± 0.6420 kg/pig, and overall FCR was 2.39 ± 0.350 between days 0 and 98 of the trial. The FS was, on average, 6.5 ± 0.2 for the total experimental period and was not affected by the treatments (Figure 2).

Mortality rate was 2.1% for the BG 1 group and 3.3% for the BG 2 group. No mortality was recorded in the CON group. Diets did not interfere with the daily frequency of the visits to the feed stations. However, the ratio of real feed consumption during the visits was significantly (*p* < 0.01) decreased when the pigs were fed the BG2 diet after day 50 (ranging from 95% to 97%). These effects were not observed with control or BG1 diets and were not gender-related (Figure 3).

No remarkable dietary effect was observed on production performance, except for a significant decrease (−11.6%) in the ADG of the pigs fed the BG2 diet compared with pigs fed the BG1 diet from 50 to 80 kg. However, the BG1 diet did not affect production performance when compared to the Control diet. During the last feeding phase (80–120 kg), the ADG (+14.3%) and ADFI (+5.3%) were significantly higher in boars than in gilts, regardless of the diet, and this effect was also observed for the ADG when considering the complete feeding period (+ 6.5%). Furthermore, the FCR was significantly lower in boars than in gilts, regardless of the diet, during the last feeding phase (80–120 kg; −8.3%) and the total feeding period (25–120 kg; −4.3%) (Figure 4). Likewise, estimated feed costs were not affected by the tested diets, with a significant gender effect in the finisher period, where the costs to produce boars were 5.5% higher than those used to grow (Table 2).

### 3.2. Carcass Characteristics

At the start of the trial, average muscle thickness was 31.3 mm, reaching 52.7 mm at the end of the trial without differences among treatments. Individual slaughter data were obtained for all grower-finisher pigs in the trial, except those that had died. Dietary treatments did not affect the carcass parameters. Backfat thickness was significantly lower in gilts (−6.8%), whereas muscle depth (+4.7%) and lean meat (+0.9%) were significantly higher in gilts than in boars (Table 3).

### 3.3. Serum Levels of Urea, Insulin, and Leptin

No differences were observed when comparing the serum levels of urea, insulin, and leptin. Serum levels of urea, insulin, and leptin were significantly higher in gilts than in boars (Table 4).

## 4. Discussion

Based on our results, dietary inclusion of BG (6.5–13 g/kg) did not affect production performance and carcass quality of growing-finisher pigs, except for a decrease in the ADG when 50–80 kg pigs were fed a diet with the highest BG inclusion. Due to its bitter taste, caused by the presence of compounds such as saponins [19], an impaired performance could be expected. However, the decreased ADG was not related to a decreased feed intake. The BG2 diet may have decreased palatability over time because after day 50, the pigs were visiting the feed stations with the same frequency compared to the other groups, but 3–5% of these visits did not result in feed intake. The decrease in ADG was not observed in the pigs of 80–120 kg and affected neither the overall ADG nor the FCR. Dietary BG extract does not decrease body weight and, although insulin stimulates the glucose transport to skeletal muscle tissue [20], no effect on muscle thickness was observed in the present study. Climatic and agriculture conditions in Europe are not optimal for the cultivation of crops like soybeans [21]. These same authors also pointed out the Chinese dependency on the international supply of soybeans. Hence, European and Chinese animal feed are not only corn-soybean-meal-based, but usually contain a large number of ingredients at variable inclusion levels (ranging from 0.5 to ~45%). In the present trial, the possibility to add an alternative ingredient at a 0.65% inclusion level represents a promising solution to include vegetable leftovers in the diet while keeping animal production performance like that obtained with conventional ingredients.

Bitter gourd leaves are rich in phenolic compounds (catechins), cucurbitacin, saponins, and alkaloids [19], which are known because of their antioxidant capacity and ability to inhibit lipid peroxidation, but without a clear effect on carcass quality in pigs [22]. In the present study, no dietary effect was observed on carcass quality and muscle thickness. The pigs were fed commercial diets following the nutrient requirements and were not subjected to any challenging or stressful conditions. There are no available studies applying these leftovers for animal production. In general, the functional food properties are studied in laboratory animals [9,10,11,12,13,14] as models for human. Therefore, it is not possible to perform a critical comparison with other studies.

The absence of differences among the diets was also confirmed by the serum levels of insulin, leptin, and urea. As mentioned above, pigs were not subjected to any dietary challenges that could have stimulated a rise in serum glucose levels. Serum was sampled approximately 14 weeks after the start of the trial and 8 h after fasting. Probably, serial blood sampling during the feeding trial, especially when the pigs weighed 50–80 kg, could provide more information. In a previous study in pigs [20] challenged with poor hygiene conditions, insulin plasma levels were higher in those with a better feed efficiency. These authors recorded a first insulin peak after 45 min fasting and compared the plasma insulin levels up to 4 h after fasting. In the present trial, the absence of an anti-lipolytic effect of the BG diets was confirmed by the unchanged backfat thickness and leptin serum levels. Leptin is a hormone mostly secreted by adipocytes, playing a role in several functions such as energy balance and appetite regulation [23], and it is positively correlated with backfat thickness [24]. The experimental diets did not affect the catabolism of amino acids at the end of the trial, as demonstrated by the unchanged levels of blood urea nitrogen. Insulin sensitivity is negatively related to plasma urea [25]. This corroborates with the results related to body weight, which was similar among the treatments, and the unaltered insulin plasma levels. The highest inclusion level of BG in the present study was 13 g/kg diet (1.3%), and most likely, higher BG inclusion levels are necessary to observe effects on insulin homeostasis and protein metabolism. In a previous study, we observed that a 4% inclusion level of BG stems and leaves increased plasma insulin concentrations in female adult minipigs (data not published). However, it is still necessary to evaluate such a high inclusion in the diets of growing-finish pigs and the effects on the nutritional composition of the diet.

The absence of negative impacts on production performance and carcass quality shows that supplementing diets with 6.5 g/kg BG is promising. However, the costs are still high due to the small-scale procedure used to dry the BG leftovers (approximately 0.80 EUR/kg BG). To become economically attractive, the costs should be decreased to at least 0.35 EUR/kg BG. Drying costs are not the only bottleneck; in addition, the use of fossil-fuelled systems should be avoided to maintain BG as a sustainable alternative. Processing costs and energy sources are a concern in the production of alternative ingredients [5]. An increased cost to produce animal protein reflects the ineffective use of nutrients and excretion, e.g., of nitrogen, to the environment, as well as an increase in carbon emission [26]. The introduction of vegetable leftovers in pig feed is unavoidable due to upcoming food-feed competition. As for any other feed ingredient, the nutrition value and quality of the leftovers may vary with plant genotype, seasonality, storage condition, and processing technique [6]. The application of vegetable leftovers in animal feed is still far to be accepted in many countries and more dissemination of recent findings and technologies is needed to implement this practice [27].

## 5. Conclusions

This study shows that BG stems and leaves can partially (6.5 g/kg) replace conventional ingredients in swine diets without interfering with pig production performance and carcass quality. However, the processing costs should be decreased to make this approach an economically and environmentally attractive one.

## Figures and Tables

**Figure 1 animals-13-02159-f001:**
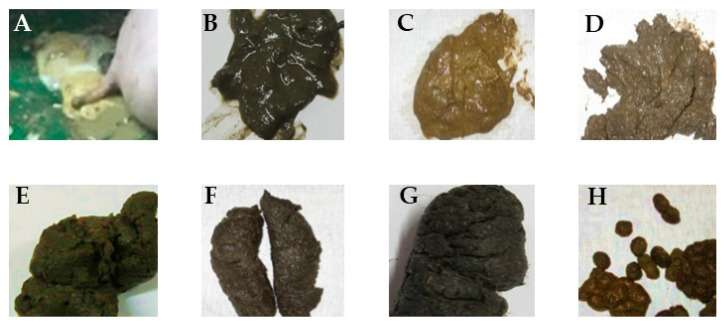
Faecal scores (FSs) for consistency. The FS given for consistency ranged from 2 to 9. The FSs 1 and 10 represent water diarrhoea and no faeces present, respectively. The scores were given as: (**A**): FS2 is characterised by severe thin diarrhoea, which flows through slatted floor; (**B**): FS3 is characterised by thin diarrhoea; (**C**): FS4 is characterised by a pudding-shaped consistency; (**D**): FS5 is characterised by soft shapeless consistency; (**E**): FS6 is characterised by a solid dropping consistency without structure; (**F**): FS7 is firm and shaped; (**G**): FS8 is characterised by a firm and cracked shape; (**H**): FS9 is characterised by hard, dry, and lumpy structures that fall apart after shaking.

**Figure 2 animals-13-02159-f002:**
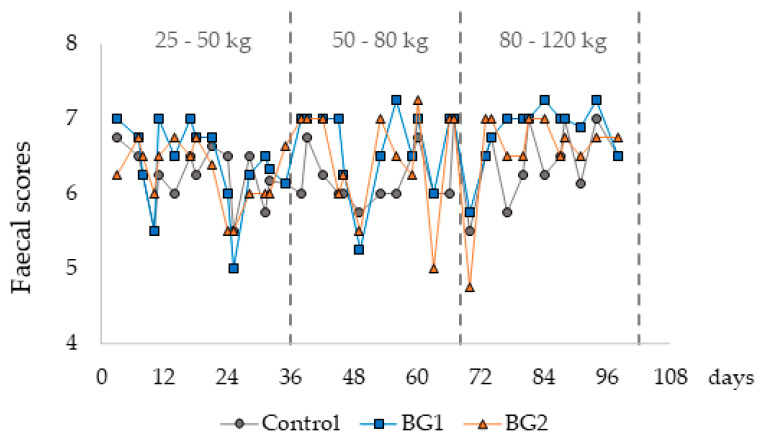
Mean faecal scores within each experimental diet. BG1: bitter gourd at an inclusion level of 6.5 g/kg diet. BG2: bitter gourd at an inclusion level of 13 g/kg diet.

**Figure 3 animals-13-02159-f003:**
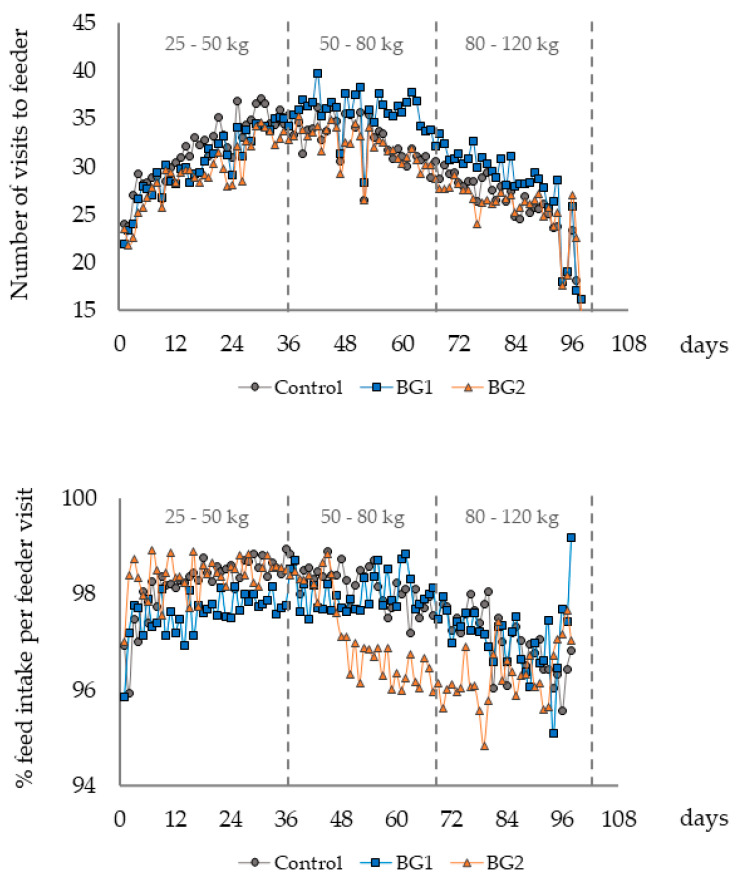
Mean number of visits to individual feeding station and percentage of visits combined with feed intake per day within each experimental diet. BG1: bitter gourd at an inclusion level of 6.5 g/kg diet. BG2: bitter gourd at an inclusion level of 13 g/kg diet.

**Figure 4 animals-13-02159-f004:**
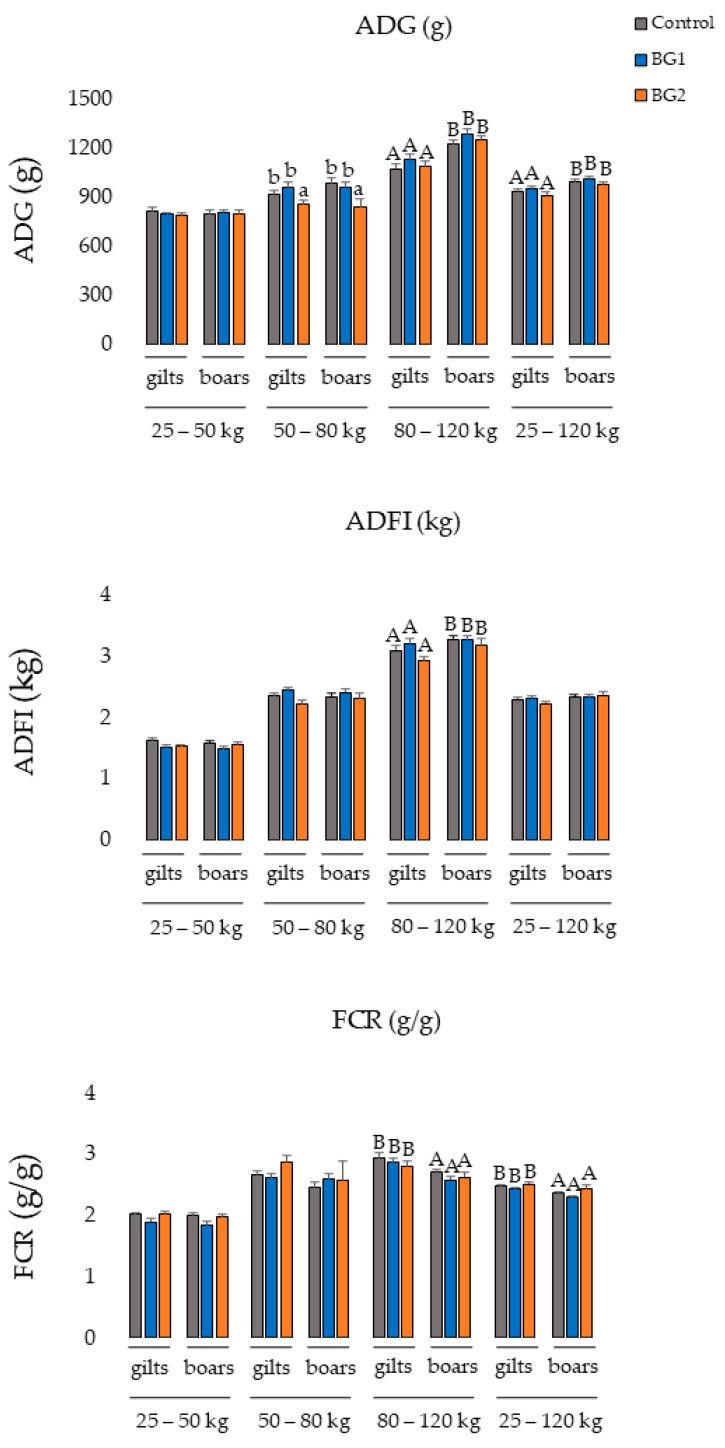
Mean (±SEM) average daily gain (ADG), average daily feed intake (ADFI), and feed conversion ratio (FCR) during the different feeding periods and the overall feeding period. Different lower-case superscripts (a, b) indicate significant differences among diets (*p* < 0.05). Different upper-case superscripts (A, B) indicate significant differences between gilts and boars fed the same diet (*p* < 0.05). BG1: bitter gourd at an inclusion level of 6.5 g/kg diet. BG2: bitter gourd at an inclusion level of 13 g/kg diet.

**Table 1 animals-13-02159-t001:** Experimental treatments and diets.

Treatments	Dosage Bitter Gourd (BG) in g/kg			
	25–35 kg	35–50 kg	50–80 kg	80–120 kg
CON				
BG1	6.25	6.25	6.80	6.50
BG2	6.25	12.50	13.60	13.00

CON: control. BG1: bitter gourd at an inclusion level of 6.5 g/kg diet. BG2: bitter gourd at an inclusion level of ~13 g/kg diet.

**Table 2 animals-13-02159-t002:** Estimation ^1^ of average feed costs (EUR) per pig in the different periods.

Treatments	Gender	25–50 kg		50–80 kg		80–120 kg		25–120 kg	
CON	Gilts	25.78		24.68		38.02		88.48	
BG1	Gilts	24.11		25.95		39.31		89.37	
BG2	Gilts	25.09		24.35		39.89		89.33	
CON	Boars	25.02		24.57		40.12		89.71	
BG1	Boars	23.84		25.39		40.92		90.16	
BG2	Boars	25.69		25.63		42.61		93.94	

CON		25.40		24.62		39.07		89.09	
BG1		23.98		25.67		40.12		89.76	
BG2		25.39		24.99		41.25		91.64	

	Gilts	24.99		24.99		39.07	a	89.06	
	Boars	24.85		25.19		41.22	b	91.27	

		*p*-value	SEM	*p*-value	SEM	*p*-value	SEM	*p*-value	SEM
Treat × Gender		0.70	1.04	0.18	0.70	0.87	1.61	0.40	2.11
Treatment		0.53	1.00	0.48	0.60	0.58	1.43	0.61	1.81
Gender		0.31	0.26	0.63	0.30	0.02	0.61	0.08	0.88

^1^ Estimation was based on the current market price of feedstuffs in Europe (25–50 kg: 0.403 EUR per kg feed; 50–80 kg: 0.390 EUR per kg feed; 50–80 kg: 0.391 EUR per kg feed). The costs for drying the bitter gourd leftovers were approximately 0.80 EUR per kg feedstuff. Different lower-case (a, b) indicate significant differences within the same column (*p* < 0.05). SEM: standard error of the means. CON: Control. BG1: bitter gourd at an inclusion level of 6.5 g/kg diet. BG2: bitter gourd at an inclusion level of 13 g/kg diet.

**Table 3 animals-13-02159-t003:** Results for average carcass weight (CW), dressing, backfat thickness (BF), muscle thickness (MT), and lean meat (LM) at slaughter.

Treatments	Gender	CW (kg)		Dressing (%)		BF (mm)		MT (mm)		LM (%)	
CON	Gilts	93.20		76.56		10.43		64.78		61.37	
BG1	Gilts	93.36		76.75		10.49		67.54		61.39	
BG2	Gilts	93.17		76.62		10.51		67.79		61.38	
CON	Boars	92.27		75.76		11.69		63.50		60.51	
BG1	Boars	92.89		76.45		10.91		63.78		61.03	
BG2	Boars	93.44		76.75		11.14		63.90		60.89	

CON		92.74		76.16		11.06		64.14		60.94	
BG1		93.12		76.60		10.70		65.66		61.21	
BG2		93.31		76.68		10.82		65.84		61.13	

	Gilts	93.24		76.64		10.48 a		66.70 b		61.38 b	
	Boars	92.87		76.32		11.25 b		63.73 a		60.81 a	

		*p*-value	SEM	*p*-value	SEM	*p*-value	SEM	*p*-value	SEM	*p*-value	SEM
Treat × Gender		0.77	0.86	0.80	0.73	0.40	0.39	0.32	1.12	0.46	0.25
Treatment		0.80	0.63	0.76	0.54	0.73	0.32	0.35	0.88	0.65	0.21
Gender		0.61	0.50	0.60	0.41	0.01	0.19	<0.001	0.58	<0.01	0.13

Different lower-case letters (a,b) indicate significant differences within the same column (*p* < 0.05). CON: Control. BG1: bitter gourd at an inclusion level of 6.5 g/kg diet. BG2: bitter gourd at an inclusion level of 13 g/kg diet.

**Table 4 animals-13-02159-t004:** Mean serum levels of urea (mmol/L), insulin (µU/mL), and leptin (ng/mL) in pigs fed the experimental diets.

Treatments	Gender	Urea (mmol/L)		Insulin (µU/mL)		Leptin (ng/mL)	
CON	Gilts	3.15		3.19		0.99	
BG1	Gilts	3.63		3.31		1.80	
BG2	Gilts	3.23		3.76		1.29	
CON	Boars	2.99		1.74		0.01	
BG1	Boars	3.16		2.03		0.05	
BG2	Boars	2.11		2.50		0.03	

CON		3.07		2.47		0.50	
BG1		3.39		2.67		0.92	
BG2		2.67		3.13		0.66	

	Gilts	3.33	b	3.42	b	1.36	b
	Boars	2.75	a	2.09	a	0.03	a

		*p*-value	SEM	*p*-value	SEM	*p*-value	SEM
Treat × Gender		0.20	0.34	0.97	0.79	0.16	0.20
Treatment		0.24	0.28	0.64	0.66	0.17	0.15
Gender		0.03	0.16	0.03	0.36	<0.001	0.11

Different lower-case letters (a, b) indicate significant differences within the same column (*p* < 0.05). CON: Control. BG1: bitter gourd at an inclusion level of 6.5 g/kg diet. BG2: bitter gourd at an inclusion level of 13 g/kg diet.

## Data Availability

The original contributions presented in the study are included in the article/Appendix A; further inquiries can be directed to the corresponding authors.

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
