# Peer review of "Effects of the Inclusion of Dietary Bitter Gourd (Momordica charantia) on the Performance and Carcass Characteristics of Pigs: Potential Application in the Feed Chain"

_animals, 2023, doi:10.3390/ani13132159_

Round 1
Reviewer 1 Report
Since this ms presents a alternative way for pig feed stuff, I am afraid the idea be not meaningful to readers. Firstly, supplement amount is only 0.6-1.3%, which most likely seems to be used as additives, but this additives offer no advantage to growth or carcass quality. If taken as feed stuff, such a small amount of supplement is not meaningful to replace feed material. Thus, I can not suggest this ms be published in Animals
This paper provided evidence that addition of 13g/kg bitter gourd leftovers in pig diet did not affect pig performance, carcass characteristics. No side or beneficial effects.
The paper adds information on using bitter gourd leftovers as feed stuff, may not invaluable, but not make sense to this field.
Previous studies pointed to extract of bitter gourd leftovers may affect insulin and glucose metabolism. This paper did not show any beneficial of adding bitter gourd leftovers in the described dose. It is hard to say what dose it add to this area.
This trail was well designed, and methods were well used.
The conclusions were drawn from their results, and do address their idea that addition of less than 1.3% bitter gourd leftovers did not affect pig performance, carcass characteristics. Unfortunately, dose not make sense.
Author Response
Since this ms presents a alternative way for pig feed stuff, I am afraid the idea be not meaningful to readers. Firstly, supplement amount is only 0.6-1.3%, which most likely seems to be used as additives, but this additives offer no advantage to growth or carcass quality. If taken as feed stuff, such a small amount of supplement is not meaningful to replace feed material. Thus, I can not suggest this ms be published in Animals.
A: In the American continent is very common to use corn and soybean as the main feedstuffs in animal diet. As a result, they are used at high inclusion levels. In Europe and Asian countries, the dietary composition contains several ingredients at inclusion levels ranging from 0.5 to 45% as shown in the dietary composition of the present study. Please, see supplementary Table 1. We agree with the present reviewer that the present amount will not replace a major number of feedstuffs but will help to decrease the use of ingredients that are also shared in the preparation of food. Furthermore, supports circular economy applying vegetable leftovers in animal feed. This comment was added in the Discussion section (L262-269; L311-316).
This paper provided evidence that addition of 13g/kg bitter gourd leftovers in pig diet did not affect pig performance, carcass characteristics. No side or beneficial effects. The paper adds information on using bitter gourd leftovers as feed stuff, may not invaluable, but not make sense to this field.
A: We must disagree with the present reviewer. Actually, it makes sense as an alternative ingredient to be added in the pig diet, as well as supports circularity because leftovers are produced daily at large amounts but just wasted or used for biomass instead of being transformed into more valuable products. This information is now included in the manuscript to properly inform the readers (L49-54).
Previous studies pointed to extract of bitter gourd leftovers may affect insulin and glucose metabolism. This paper did not show any beneficial of adding bitter gourd leftovers in the described dose. It is hard to say what dose it add to this area.
A: The fact that bitter gourd leftovers are not harming production performance and can be added to pig diet is already an advantage for circularity.
This trail was well designed, and methods were well used.
A: We acknowledge the present comment.
The conclusions were drawn from their results and do address their idea that addition of less than 1.3% bitter gourd leftovers did not affect pig performance, carcass characteristics. Unfortunately, dose not make sense.
A: Please, read the revised version of the present manuscript.
Reviewer 2 Report
The authors studied the potential of using BG leftovers as a replacement for traditional feed ingredients to reduce the loss and make the industry more eco-friendly. From the data in growing animals across the 3 phases in both sexes, the authors found that BG addition at 6.5g/kg is a promising strategy for such replacement without negative impact on growth performance or carcass quality, though the cost for BG processing could be improved to further reduce the cost. Overall, the manuscript is well written, with the content highly relevant to industry. Following issues should be revised/taken into consideration before publication:
- Please mention the details in methods part regarding the source of BG and the processing of the leftovers. The nutrition value and quality of BG leftover, as well as the processing quality can vary a lot by season, storage condition, and techniques, etc. This should be brought into consideration and discussed.
- In the introduction (line 80), you stated that in the rat study, up to 4g/kg BG chronic exposure lead to nephrotoxic effect with elevated serum urea levels. Why in this study you chose even higher doses? How did you come up with the doses in this study?
- Method 2.4 (line 117), for the measurement of fecal consistency, you should list all the details regarding the score system as there is no universal or golden rule so far for such procedures. A representative picture for each score would be highly suggested. Also, you should mention if the scoring was always done by the same experimenter(s) and if this is blinded. This is also to ensure the consistency if other researchers want to use the same method for future studies. Again, since it is a visual measurement method, a representative picture of faeces from each group for comparison would be of great help.
- Results 3.1 (line 165), the score should be shown in a graph at different experimental time points within groups (i.e. 3 dotted lines) for clear comparisons as you did for number of visits even there is no difference.
- Figure 2 (line 186): Are you sure the FCR in gilts and boars are different in all feeding groups when comparing throughout the whole growing phases (25-120 kg)? The error bars in boars group are high and the height of the means between sexes are similar, in other words, the mean+/-SEM of gilts are well included within that from the boars in all feeding groups. Please double check and specify the statistical method you used for comparison.
Author Response
The authors studied the potential of using BG leftovers as a replacement for traditional feed ingredients to reduce the loss and make the industry more eco-friendly. From the data in growing animals across the 3 phases in both sexes, the authors found that BG addition at 6.5g/kg is a promising strategy for such replacement without negative impact on growth performance or carcass quality, though the cost for BG processing could be improved to further reduce the cost. Overall, the manuscript is well written, with the content highly relevant to industry.
A: We would like to acknowledge the pertinent suggestions and comments from the present reviewer. All the changes were performed and helped to improve the quality of this manuscript.
- Please mention the details in methods part regarding the source of BG and the processing of the leftovers. The nutrition value and quality of BG leftover, as well as the processing quality can vary a lot by season, storage condition, and techniques, etc. This should be brought into consideration and discussed.
A: The processing of the leftovers is now described in item 2.2 (L100-107). In the discussion section it is now pointed out the possible quality variation in these leftovers (L311-316).
- In the introduction (line 80), you stated that in the rat study, up to 4g/kg BG chronic exposure lead to nephrotoxic effect with elevated serum urea levels. Why in this study you chose even higher doses? How did you come up with the doses in this study?
A: We apologize for the misunderstanding. The mice were exposed via intraperitoneal administration to 4 g/kg body weight. In another study, toxic effects were observed when rats were submitted to oral administration of 2 g/kg bw. Therefore, 13 g of leftovers for a 25 kg bw pigs was expected to be in the safe side, as observed in the present trial. The text was corrected and the study in rats was added to clarify this information (L68-72).
- Method 2.4 (line 117), for the measurement of fecal consistency, you should list all the details regarding the score system as there is no universal or golden rule so far for such procedures. A representative picture for each score would be highly suggested. Also, you should mention if the scoring was always done by the same experimenter(s) and if this is blinded. This is also to ensure the consistency if other researchers want to use the same method for future studies. Again, since it is a visual measurement method, a representative picture of faeces from each group for comparison would be of great help.
A: We added representative images (Figure 1) and a reference related to this scoring system.
- Results 3.1 (line 165), the score should be shown in a graph at different experimental time points within groups (i.e. 3 dotted lines) for clear comparisons as you did for number of visits even there is no difference.
A: The dotted lines were added. Figure 2 now gives faecal scores, and previous Figure 1 is now Figure 3.
- Figure 2 (line 186): Are you sure the FCR in gilts and boars are different in all feeding groups when comparing throughout the whole growing phases (25-120 kg)? The error bars in boars group are high and the height of the means between sexes are similar, in other words, the mean+/-SEM of gilts are well included within that from the boars in all feeding groups. Please double check and specify the statistical method you used for comparison.
A: This is now Figure 4. We have observed that the error bars were not correct, and the graph was revised. The differences are correct. Please, accept our apologies.
Reviewer 3 Report
In this study, Guan et al aimed to evaluate diets with or without bitter gourd (BG) leftovers on feed costs, growth performance, carcass traits, and serum levels of urea, insulin, and leptin in growing-finishing pigs. BG leftovers may be an alternative feed source, however, there are several significant problems with this study:
1. How to get BG leftovers in this study? It should be mentioned in Materials and Methods.
2. Why did the author measure the faecal consistency? The results of this indicator was also not present in Results.
3. How to detect the muscle and fat thickness? It should be described clearly.
4. What’s the meaning of IVOG in Figure 1. Furthermore, the three groups in Figure 1 were not easy to distinguish. Three different colors could be used to separate different groups.
5. Glucose levels should be measured in Table 4.
6. The table title of Table 4 was inappropriate.
Author Response
Reviewer 3
- How to get BG leftovers in this study? It should be mentioned in Materials and Methods.
A: Details are now given (L100-107).
- Why did the author measure the faecal consistency? The results of this indicator was also not present in Results.
A: This is a regular measurement performed in pig trials to monitor diarrhea. There were no differences in the faecal scores, but the information was present in L193-194.
- How to detect the muscle and fat thickness? It should be described clearly.
A: This is now described and a reference is given (L154-157].
- What’s the meaning of IVOG in Figure 1. Furthermore, the three groups in Figure 1 were not easy to distinguish. Three different colors could be used to separate different groups.
A: This is an abbreviation for the individual feeder system. We changed this abbreviation in the manuscript by feeder. Now the Figures are in colour.
- Glucose levels should be measured in Table 4.
A: Unfortunately, we were not able to measure serum glucose levels.
- The table title of Table 4 was inappropriate
A: It is now corrected. Our apologies for the mistake.
Reviewer 4 Report
Report on the manuscript animals-2376760 entitled: Effects of the inclusion of dietary bitter gourd (Momordica charantia) on the performance and carcass characteristics of growing finishing pigs: potential application in the feed chain.
- I cannot find Table 1 or Table S1 anywhere.
- The title must be rewritten. The dietary intervention was carried out from the starter (25 kg according to the authors) to the finishing stage.
Hard to review without Tables 1 and S1.
Either way, growing-finishing must be hyphenated.
- Were “boars” actually used in this study? Boars as “non-castrated” males? Or were “barrows” actually used?
- L. 128-130. Source of the equation?
- L. 172-181. Please, add the % of the difference to the description of Fig. 2 and Table 2 results.
- L. 198-202. Please, add the % of difference when statistically significant differences were detected.
- L. 208-210. Gender P < 0.05 not described.
- L. 211. The Table 4 caption is wrong.
- It is obvious that the SD bars in Figure 2 have been crudely drawn by hand. This leads to a lack of confidence in the results described...
- The discussion is poor and needs to be improved. It is not acceptable to just write the number of the reference at the end of a sentence. A critical discussion must be included.
In addition, a critical comparison to the results from manuscripts that consider similar additives should be considered.
- L. 22. Delete "of" before 25.
- L. 34. "...the pen".
- L. 44. "...to producing..."
Please, review the whole manuscript. Consider a native speaker.
Author Response
- I cannot find Table 1 or Table S1 anywhere.
A: Please, accept our apologies for this. The Tables are now properly uploaded.
- The title must be rewritten. The dietary intervention was carried out from the starter (25 kg according to the authors) to the finishing stage.
A: We removed the “growing-finishing” indication in the title.
Hard to review without Tables 1 and S1.
A: Please, see added tables.
Either way, growing-finishing must be hyphenated.
A: It was not hyphenated only in the Title. This is now deleted.
- Were “boars” actually used in this study? Boars as “non-castrated” males? Or were “barrows” actually used?
A: Yes, this is now clarified (L93).
- L. 128-130. Source of the equation?
A: A reference is now given (L154-157).
- L. 172-181. Please, add the % of the difference to the description of Fig. 2 and Table 2 results.
A: Percentages are now described.
- L. 198-202. Please, add the % of difference when statistically significant differences were detected.
A: Percentages are now described.
- L. 208-210. Gender P < 0.05 not described.
A: This difference is now described.
- L. 211. The Table 4 caption is wrong.
A: It is now corrected.
- It is obvious that the SD bars in Figure 2 have been crudely drawn by hand. This leads to a lack of confidence in the results described...
A: We have observed that the error bars were not correct, and the graph was revised. They were not drawn by hand, but the quality of the Figure was not good. See new Figure (now Figure 4).
- The discussion is poor and needs to be improved. It is not acceptable to just write the number of the reference at the end of a sentence. A critical discussion must be included.
In addition, a critical comparison to the results from manuscripts that consider similar additives should be considered.
A: The Discussion section was improved. We also explain in the Discussion section that bitter gourd is not an additive, but a feed ingredient (L262-269). Unfortunately, there are no other studies evaluating the effect of bitter gourd on animal production.
Comments on the Quality of English Language
A: This manuscript was edited by a British professional company
- L. 22. Delete "of" before 25.
A: Deleted
- L. 34. "...the pen".
A: Done.
- L. 44. "...to producing..."
A: Done.
Round 2
Reviewer 1 Report
I still insist on that this ms dose not provide significant informaiton to pig feed stuff replacement. However, this test was well desinged, and results were correctely anaysed. Considering it may add informaiton on understanding of bitter gourd when it was added in pig feed, and the auther has made revisions properly, I suggest it be accepted.
Author Response
R: I still insist on that this ms dose not provide significant informaiton to pig feed stuff replacement. However, this test was well desinged, and results were correctely anaysed. Considering it may add informaiton on understanding of bitter gourd when it was added in pig feed, and the auther has made revisions properly, I suggest it be accepted.
A: We acknowledge the decision of the present reviewer, and we are confident that this study will help other research groups focused on circularity.
Reviewer 4 Report
The authors must consider:
- Fig. 2. and Fig. 3. Please, describe what Control, BG1, and BG2 mean.
- Fig. 4. The statistical analysis must be reviewed.
ADG 50-80, boars control should have a “b” not an “ab” which belongs to boars BG1.
According to the error bars, ADFI gilt 80-120 should be statistically different from the others.
According to the error bars, FCR BG1 of gilts and boars 25-50 and gilts 50-80 should be statistically different from the others.
- Table 2 and Table 3. It is impossible to get such SEM values. Please, review.
- Discussion must be improved. It is still unspecific.
--
Author Response
Fig. 2. and Fig. 3. Please, describe what Control, BG1, and BG2 mean.
A: This information added as footnote is all related Figures and Tables.
Fig. 4. The statistical analysis must be reviewed.
A: The statistical analysis was performed by a professional statistician and all data was again double checked. All error bars and SEM are also correct. However, to fulfill the request of the present reviewer, we removed the Tukey test from the analysis and left only Fisher’s least significant difference (LSD) test. Based on this, some letters were changed.
ADG 50-80, boars control should have a “b” not an “ab” which belongs to boars BG1.
A: By excluding Tukey test from the analysis, now we have a “b” instead of “ab”.
According to the error bars, ADFI gilt 80-120 should be statistically different from the others.
A: The analysis always considered interaction diet x gender and not as separate blocks per gender. Moreover, the p-value is 0.243. Therefore, we cannot put a letter indicating a statistical difference.
According to the error bars, FCR BG1 of gilts and boars 25-50 and gilts 50-80 should be statistically different from the others.
A: The statistical analysis did not include age and weight of the pigs because such a change is already expected and not related to the experimental design. This is now clarified in the “Statistical analysis” item. (L184-200).
Table 2 and Table 3. It is impossible to get such SEM values. Please, review.
A: The SEM values are possible and correct. As an example, we add here part of output from Table 3, as obtained from the Genstat program:
For CW (kg)
Standard errors of means
Table Sexe Trt Sexe
Trt
rep. 120 80 40
e.s.e. 0.50 0.63 0.86
d.f. 170 13 44.13
Except when comparing means with the same level(s) of
Trt 0.84
d.f. 170
For Dressing (%)
Standard errors of means
Table Sexe Trt Sexe
Trt
rep. 120 80 40
e.s.e. 0.412 0.537 0.725
d.f. 170 13 41.86
Except when comparing means with the same level(s) of
Trt 0.694
d.f. 170
For BF (mm)
Standard errors of means
Table Sexe Trt Sexe
Trt
rep. 120 80 40
e.s.e. 0.19 0.32 0.39
d.f. 170 13 29.33
Except when comparing means with the same level(s) of
Trt 0.33
d.f. 170
For MT (mm)
Standard errors of means
Table Sexe Trt Sexe
Trt
rep. 120 80 40
e.s.e. 0.58 0.88 1.12
d.f. 170 13 33.29
Except when comparing means with the same level(s) of
Trt 0.98
d.f. 170
For LM (%)
Standard errors of means
Table Sexe Trt Sexe
Trt
rep. 120 80 40
e.s.e. 0.13 0.20 0.25
d.f. 170 13 30.42
Except when comparing means with the same level(s) of
Trt 0.21
d.f. 170
Discussion must be improved. It is still unspecific.
A: It is quite difficult for us to understand what must be added in the Discussion section without a specific remark. As previously informed, there are no other studies evaluating the effect of bitter gourd on animal production. Furthermore, comparing bitter gourd with other vegetable leftovers, which contain a completely different chemical composition will not bring additional information to the present findings.

Round 3
Reviewer 4 Report
--
--